# Tumor Signature Analysis Implicates Hereditary Cancer Genes in Endometrial Cancer Development

**DOI:** 10.3390/cancers13081762

**Published:** 2021-04-07

**Authors:** Olga Kondrashova, Jannah Shamsani, Tracy A. O’Mara, Felicity Newell, Amy E. McCart Reed, Sunil R. Lakhani, Judy Kirk, John V. Pearson, Nicola Waddell, Amanda B. Spurdle

**Affiliations:** 1Department of Genetics and Computational Biology, QIMR Berghofer Medical Research Institute, Brisbane 4006, Australia; olga.kondrashova@qimrberghofer.edu.au (O.K.); jannah@genieus.co (J.S.); Tracy.OMara@qimrberghofer.edu.au (T.A.O.); Felicity.Newell@qimrberghofer.edu.au (F.N.); John.Pearson@qimrberghofer.edu.au (J.V.P.); Nic.Waddell@qimrberghofer.edu.au (N.W.); 2UQ Centre for Clinical Research, Faculty of Medicine, The University of Queensland, Brisbane 4029, Australia; amy.reed@uq.edu.au (A.E.M.R.); s.lakhani@uq.edu.au (S.R.L.); 3Anatomical Pathology, Pathology Queensland, Brisbane 4029, Australia; 4Familial Cancer Service, Crown Princess Mary Cancer Centre, Westmead Hospital, Sydney 2145, Australia; judy.kirk@sydney.edu.au; 5Centre for Cancer Research, The Westmead Institute for Medical Research, Sydney Medical School, University of Sydney, Sydney 2145, Australia

**Keywords:** endometrial cancer, genomic sequencing, tumor mutational signatures, hereditary cancer genes, mismatch repair, familial cancer

## Abstract

**Simple Summary:**

Women with a family history of cancer are at increased risk of cancer, including endometrial cancer (affecting the womb lining). In some of the women with such family history, the risk can be explained by deleterious changes in mismatch repair genes that cause Lynch syndrome. This study explored the role of other genes in risk of endometrial cancer, using several approaches. The number and type of changes in gene sequence information in women with endometrial cancer was compared to that from individuals in the general population. Gene sequence changes in endometrial cancer patients with a family history of cancer were also analyzed. Lastly, endometrial cancers from individuals with gene changes were examined for distinctive genomic patterns expected to be seen if a gene change was driving the cancer. This study has identified several additional genes for further exploration in relation to endometrial cancer risk and therapy.

**Abstract:**

Risk of endometrial cancer (EC) is increased ~2-fold for women with a family history of cancer, partly due to inherited pathogenic variants in mismatch repair (MMR) genes. We explored the role of additional genes as explanation for familial EC presentation by investigating germline and EC tumor sequence data from The Cancer Genome Atlas (*n* = 539; 308 European ancestry), and germline data from 33 suspected familial European ancestry EC patients demonstrating immunohistochemistry-detected tumor MMR proficiency. Germline variants in MMR and 26 other known/candidate EC risk genes were annotated for pathogenicity in the two EC datasets, and also for European ancestry individuals from gnomAD as a population reference set (*n* = 59,095). Ancestry-matched case–control comparisons of germline variant frequency and/or sequence data from suspected familial EC cases highlighted *ATM*, *PALB2*, *RAD51C*, *MUTYH* and *NBN* as candidates for large-scale risk association studies. Tumor mutational signature analysis identified a microsatellite-high signature for all cases with a germline pathogenic MMR gene variant. Signature analysis also indicated that germline loss-of-function variants in homologous recombination (*BRCA1*, *PALB2*, *RAD51C*) or base excision (*NTHL1*, *MUTYH*) repair genes can contribute to EC development in some individuals with germline variants in these genes. These findings have implications for expanded therapeutic options for EC cases.

## 1. Introduction

Endometrial cancer (EC) is the most commonly diagnosed gynecological malignancy, with an increased prevalence rate in developed countries [1]. Modifiable factors such as obesity, lifestyle, and hormone levels are associated with increased risk of EC, and women with a family history of EC or other cancers, such as colorectal, are at ~2–3 fold increased risk of EC [2]. The genetic factors identified to date are either common low-risk cancer predisposition variants that act together to cause polygenic disease, or rare high-risk pathogenic variants in cancer syndrome genes generally present in patients with a monogenic disease phenotype [3].

The major known monogenic form of EC is Lynch syndrome, caused by germline pathogenic variants impacting the mismatch repair (MMR) genes *MLH1*, *MSH2*, *MSH6*, *PMS2*, as well as *EPCAM* deletions, which impact *MSH2* expression. Lynch syndrome accounts for approximately 3–5% of EC at the population level and an increased proportion in cases with family history of colorectal, endometrial and other cancers [4]. The lifetime cumulative risk of EC for women with Lynch syndrome is 40–70%, depending on which MMR gene is disrupted [5]. EC is also a spectrum cancer of Cowden syndrome, caused by the inheritance of pathogenic *PTEN* variants. The cumulative risk of EC for women up to 60 years of age with Cowden syndrome is around 20% [6]. Studies to date suggest that *PTEN* pathogenic variants are very rarely detected in the general population, and mostly in the context of clinical features of Cowden syndrome [7].

Results from a recent study assessing risk associated with reported family history of endometrial and other cancers, after considering proband MMR proficiency and MMR germline test results, indicate that the genetic basis for a substantial fraction of familial EC patients with MMR deficient and MMR proficient tumors remains unexplained [8].

Several genes involved in other hereditary cancer syndromes have been either directly or indirectly implicated in hereditary EC, but with insufficient or conflicting support that germline DNA gene testing would provide clinically useful information for genetic counseling [4]. These include established hereditary cancer syndrome genes, such as *POLE*, *POLD1*, *MUTYH, STK11, TP53, BRCA1* and *BRCA2* [9,10,11,12,13,14,15,16,17,18,19,20,21]. Additionally, germline alterations in a number of other known or candidate cancer risk genes have been identified in EC patients from clinical or research studies, including homologous recombination (HR) DNA repair pathway genes (reviewed in [4]). However, because of the paucity of studies focusing on EC and limitations due to study design, there is uncertainty regarding EC risk associated with variants in these genes [4,22].

To explore which genes may influence the EC risk beyond the well-recognized MMR genes, we assessed the frequency of pathogenic variants in a total of 30 known or candidate EC risk genes in publicly available EC and population data. To assist with the interpretation of the EC driver status of pathogenic variants, we performed tumor mutational signature analysis. We also sequenced and analyzed the germline exomes or whole genomes of 33 EC cases with reported family history of endometrial and other cancer types with no evidence of tumor MMR deficiency. 

## 2. Materials and Methods 

### 2.1. Study Participants and Data Resources

EC cases unselected for family history were accessed from The Cancer Genome Atlas Uterine Corpus Endometrial Carcinoma study (TCGA-UCEC; *n* = 539). Germline and tumor whole exome sequencing data was used. To align with the most recent NIH genomic data sharing policy, TCGA IDs have been de-identified. For case–control variant frequency comparison, the analysis was limited to individuals of European ancestry (*n* = 308; Appendix A). Ancestry was determined from SNP arrays and classified as European or Non-European [23]. Where SNP-determined ancestry was not available, cases were selected by self-reported race. 

GnomAD r2.1.1 database was used as a control population (*n* = 15,708 genomes and *n* = 125,748 exomes). To overcome issues around population stratification for case–control comparison, we limited our analysis to individuals of European ancestry (gnomAD—Non-Finnish Europeans; *n* = 95,095).

Suspected familial EC cases were selected from the Australian National Endometrial Cancer Study (ANECS), a population-based study of epidemiological and genetic risk factors for EC. Details of the ANECS study design, including recruitment and data collection, are described in detail in previous publications [8,24,25]. Cases were selected for this study if they met all of the following criteria: the case provided detailed cancer report information in first, second and selected third degree relatives by structured questionnaire and follow-up interview [8]; the case (or for one individual—endometrial cancer affected sister) had previously demonstrated tumor MMR proficiency using immunohistochemistry [24,25]; the case had reported at least one affected relative with a cancer diagnosis (excluding skin cancer due to the significant role of environmental factors in Australia, and excluding EC after a breast cancer diagnosis due to possible confounding by tamoxifen exposure); and there was a germline DNA sample (extracted from whole blood) available for analysis. Germline sequencing was undertaken for 33 unrelated EC cases. The clinical features of the cohort are summarized in Appendix A. Participants self-reported British/Irish heritage, and/or were confirmed to have European heritage based on genetic markers. 

### 2.2. Sequencing for Suspected Familial EC Cases

Genomic DNA was extracted from blood using a salting out method. DNA samples from 6 cases were sequenced using whole exome sequencing and 27 samples were sequenced using whole genome sequencing. Exome libraries were prepared using the Nextera Rapid Capture Exome Kit (Illumina) and sequencing was performed on the NextSeq500 (Illumina) using 2 × 150 bp reads with an average read depth of 75× (Appendix A). Whole genome sequencing was performed using HiSeq X Ten (Illumina) with an average read depth of 36× (Appendix A). Tumor DNA of one ANECS EC patient (case 28) carrying a germline *MUTYH* variant was extracted from Formalin-Fixed Paraffin-Embedded (FFPE) tissue using Qiagen DNeasy Blood and Tissue kit (Qiagen, Hilden, Germany). Tumor DNA whole genome sequencing was performed using HiSeq X Ten (Illumina, San Diego, CA, USA) to an average read depth of 12×.

### 2.3. Sequence Analysis

TCGA-UCEC sequencing data were downloaded as aligned reads (BAM format) and converted to FASTQ format for processing. 

Sequencing reads were trimmed using Cutadapt (version 1.9) [26] and aligned to the reference genome (GRCh37) with BWA-MEM (version 0.7.13) [27]. Duplicate aligned reads were marked with Picard (version 1.141) (http://picard.sourceforge.net accessed on 17 November 2015) and sorted using samtools (version 1.3) [28]. Somatic and germline variants were identified by a dual calling strategy using qSNP [29] and GATK Haplotype caller [30], as previously described [31]. For the FFPE tumor sample (case 28), single nucleotide variants (SNVs) were annotated to identify overlapping reads to prevent overcalling due to DNA fragmentation from formalin fixation. SNVs with at least 5 alternate bases after removal of overlapping reads and those absent in dbSNP were kept for signature analysis. 

Germline variants were annotated using the Ensembl Variant Effect Predictor (VEP) [32], with population allele frequency based on the Exome Aggregation Consortium (ExAC-nonTCGA v3). The *in silico* predictions were annotated using VEP-plugins: REVEL [33] and MaxEntScan [34]. Variants were also annotated for variant pathogenicity as submitted to ClinVar [35], if present in this database. 

### 2.4. Variant Prioritization

Analysis was focused on rare germline variants (minor allele frequency (MAF) of less than 1% in any population in the ExAC-nonTCGA) in 30 genes of interest [4], including the four MMR genes and *EPCAM* (Appendix A). In this study, we excluded from analysis any variants in exons 9 and 11–15 of *PMS2*, due to homology with the *PMS2L* pseudogene in these regions [36]. For *POLE* and *POLD1* genes, only missense variants were considered [37].

For the gnomAD and TCGA-UCEC dataset analysis, only pathogenic or likely pathogenic ClinVar variants or predicted truncating variants (termed as likely pathogenic in this study) were considered (Appendix A). The proportion of pathogenic/likely pathogenic carriers in TCGA and gnomAD datasets was calculated by dividing the number of observed pathogenic/likely pathogenic variants by the total number of individuals sequenced for that gene. For the gnomAD dataset, the number of individuals sequenced was calculated by halving the highest allele number for each gene. 

For the familial EC dataset, variants present in three or more samples were excluded as common variants. The remaining variants were reviewed and included if they were: (i) predicted truncating variants (nonsense, frameshift indels, and splice donor or acceptor); (ii) predicted to be deleterious by *in silico* predictions using REVEL (cutoff of ≥ 0.5) or PROVEAN (cutoff of ≤ −2.5) [38]; (iii) predicted to disrupt native donor/acceptor site or create a de novo donor splice site (including synonymous) [34]; or (iv) annotated as pathogenic, likely pathogenic or uncertain significance (VUS), with supporting evidence provided, by multiple submitters in ClinVar database. All candidate variants identified in the familial EC samples were manually reviewed in the Integrated Genome Viewer (IGV) to eliminate any artefacts. Validation of the three prioritized variants was performed by Sanger sequencing. 

### 2.5. Mutation Signature Analysis

At least 100 somatic SNVs per sample were required for signature analysis. SNV mutational signature analysis was performed using deconstructSigs and the COSMIC v2 signature catalogue with the minimum signature contribution set to 15% [39]. Default settings were used for the familial EC case 28 (whole-genome sequencing) and the exome settings for the TCGA-UCEC cohort. *De novo* signature analysis was previously performed using SigProfiler [40].

TCGA-UCEC data were assessed for tumor mutation burden (TMB), microsatellite instability (MSI) status, tumor enrichment of the germline variant in question and additional somatic variations in same gene for *POLE* and MMR genes. TMB was calculated as a number of all somatic mutations divided by the coverage (Mb) of capture kit used (hg18 Nimblegen v2—26.2 Mb, SureSelect All Exon—44 Mb, Nimblegen SeqCap EZ v2.0—36.5 Mb and Nimblegen SeqCap EZ v3.0—64 Mb). The level of MSI was assessed using MSIsensor (v0.2) on tumor-normal pairs [41]. The analysis was limited to the capture-covered regions. Samples with MSI scores ≥3.5 were classified as MSI-high. Germline variants were considered enriched in tumor if the percentage of sequence reads containing a variant was ≥60% in the tumor sample. 

*POLE* somatic mutation status for TCGA-UCEC samples was determined by checking for somatic missense *POLE* mutations in exons 9–14. MMR gene somatic mutation status for TCGA-UCEC samples was assessed using the same approach as for the germline variants. *MLH1* gene methylation and *MSH2* gene deletion (copy number-based) information for TCGA-UCEC (Firehose legacy) study [42] was downloaded from cBioPortal [43,44]. *MLH1* was classified as methylated if the beta-value was >0.3. 

### 2.6. Code and Data Availability

Scripts used for TCGA and gnomAD data analysis are available on https://github.com/okon/EC_TCGA_vs_gnomAD. TCGA-UCEC data were downloaded from GDC data portal in October 2016. GnomAD variant files (r.2.1.1) were downloaded from the gnomAD portal in April 2019. ANECS sequencing data are available upon reasonable request and subject to ethics approval.

## 3. Results

### 3.1. Germline Variants in Data from Publicly Available EC Cases

We compared the frequency of germline variants between EC cases unselected for family history (TCGA-UCEC study) and the general population (gnomAD database) in a subset of 30 genes, previously highlighted as known or purported to be associated with risk of developing EC (Appendix A) [4]. Pathogenic or likely pathogenic variants were selected based on ClinVar reports or predicted protein truncating effect, as outlined in Appendix A. We did not perform formal statistical comparisons because the EC cohort size (*n* = 308) was underpowered to detect significant differences for the expected rare observations, even for MMR genes. 

A total of 19 distinct germline pathogenic or likely pathogenic variants were detected in 12 of 30 analyzed risk genes in 25 of 308 TCGA-UCEC cases (Table 1 and Appendix A), similar to previous analyses [4,45]. The carrier frequency in the EC cases compared to the gnomAD population was more than double for three of the known MMR genes—*MSH6* (1.3% vs. 0.23%), *MSH2* (0.65% vs. 0.02%) and *PMS2* (0.32% vs. 0.13%), as well as for the HR repair genes *RAD51C* (0.97% vs. 0.1%), *PALB2* (0.32% vs. 0.14%) and *NBN* (0.32% vs. 0.15%). Pathogenic or likely pathogenic variants observed for other candidate EC risk genes occurred at less than 2-fold increased frequency or were found with a lower frequency in cases versus controls, namely: *BRCA1* (0.32% vs. 0.24%), *NTHL1* (0.65% vs. 0.45%), *FAN1* (0.32% vs. 0.31%), *SEC23B* (0.32% vs. 0.33%), *MUTYH* (1.62% vs. 1.73%) and *CHEK2* (0.97% vs. 1.86%).

### 3.2. Role of Germline Variants in Driving EC Development in TCGA-UCEC Cases

We explored the potential role of germline variants in known and candidate EC risk genes in cancer development by analyzing tumor sequencing data for evidence of tumor variant enrichment and presence of mutational signatures reflective of defective DNA repair pathways (e.g., HR pathway). We assessed 46 TCGA-UCEC cases, unselected by ancestry, with pathogenic or likely pathogenic germline variants (*n* = 31 distinct variants) in the 30 prioritized genes (Appendix A). 

Three of the eight cases with pathogenic or likely pathogenic germline variants in MMR genes had evidence of variant enrichment in tumor (one *MSH2* and two *MSH6* variants with >60% variant reads in the tumor sample; Figure 1). In three cases with *MSH2* or *MSH6* variants (one with germline variant enrichment in tumor), we detected a second somatic hit in the respective genes (Figure 1). While we did not observe tumor variant enrichment or second hits for the other three MMR-positive cases, all eight cases had high TMB (>10 Mut/Mb) indicative of MMR deficiency and MSI detected by MSIsensor. We also observed MMR-associated mutational signatures in all eight cases by *de novo* signature analysis (over 25% contribution; eight out of eight cases; Appendix A), and also by signature assignment to the 30 known COSMIC v2 signatures for two of the eight cases, further supporting the tumor driver role of MMR variants in these cases (Figure 1).

Nine cases with germline variants in HR-related genes *PALB2*, *BRCA1*, *RAD51C*, *FAN1* and *CHEK2* also showed evidence of enrichment of the germline variant in the tumor, while the other 12 cases with HR-related gene variants (seven *FAN1*, three *CHEK2*, one *BRIP1,* one *NBN*) did not (Figure 1). Using mutational signature assignment analysis, Signature 3—associated with HR deficiency, was detected in six of seven of tumors with *BRCA1, PALB2* and *RAD51C* variants. We did not observe Signature 3 in the other cases with germline alterations in HR-related genes, suggesting that they were HR pathway proficient.

One of two cases that harbored germline inactivating *NTHL1* variant (p.Gln90*) had evidence of tumor variant enrichment (Figure 1). This case showed high TMB and presence of Signature 30, characterized by the prevalence of C>T mutations and associated with deficiency in base excision repair expected due to *NTHL1* inactivation [46]. However, this case also showed high MSI and *MLH1* methylation. Finally, no cases with the germline pathogenic *MUTYH* variant (c.1187G>A, p.Gly396Asp) showed evidence of variant enrichment in the tumor nor presence of Signature 18, associated with *MUTYH* inactivation. Of note, while three cases with *MUTYH* variants had high TMB, we attributed it to MMR deficiency in the tumor due to *MLH1* methylation or deletion of *MSH2*, supported by high MSI levels and MMR-deficient mutational signatures.

### 3.3. Germline Variants in Suspected Familial EC Cases

To further explore which genes may explain the etiology of familial EC beyond the well-recognized MMR genes, we sequenced the germline exomes or whole genomes of 33 familial EC cases with no evidence of tumor MMR deficiency, and reported family history of endometrial or other cancer types. The analysis was focused on the same 30 genes as in the sections above (Appendix A). Out of the 33 cases, we identified three cases with candidate variants in the prioritized genes. These were a frameshift deletion in *PALB2*:c.3116delA (p.Asn1039Ilefs), an in-frame deletion in *ATM*:c.7638_7646del (p.Arg2547_Ser2549del) and a missense pathogenic variant *MUTYH*:c.536A>G (p.Tyr179Cys). 

The patient (case 2) carrying the pathogenic *PALB2* frameshift variant (c.3116delA, p.Asn1039Ilefs) was diagnosed with stage 1 endometrioid EC at age 70 years. She self-reported that 17 family members had been diagnosed with various types of cancer (Figure 2A), including two with EC—diagnosed at age 60 years (mother) and age 35 years (maternal aunt). Although DNA was not available from the EC-affected mother, the pedigree analysis indicates she is an obligate carrier; genotyping of three other relatives identified two carrying the *PALB2* variant, specifically a sister with colon cancer and maternal cousin with breast cancer (Figure 2A).

The in-frame deletion *ATM* variant (c.7638_7646del, p.Arg2547_Ser2549del) was predicted to be deleterious by PROVEAN and was classified as pathogenic for Ataxia-telangiectasia syndrome by multiple ClinVar submitters. The carrier (case 1) of this variant was diagnosed with stage 1 endometrioid EC at age 77 years. Two of the family members were also diagnosed with EC: mother at age 55 years and sister at age 54 years (Figure 2B). Other family members were affected with colorectal cancer at age 54 years (nephew) and cervical cancer at age 27 years (niece). DNA from relatives was not available for testing.

The missense heterozygous *MUTYH* variant (c.536A>G, p.Tyr179Cys) was identified in a female affected with grade 2 endometrioid EC at age 62 years (case 28). This *MUTYH* variant is a known common pathogenic missense variant known to cause MUTYH-associated polyposis (MAP) in Western populations when detected in homozygous or compound heterozygous state [47]. The proband reported seven family members affected with various cancers (Figure 2C), including a father diagnosed with melanoma, three relatives with breast cancer (maternal great aunt, paternal aunt, sister), two relatives affected with colorectal cancer (maternal grandfather, sister), and a maternal uncle with prostate cancer. A DNA sample was only available for the female sibling with breast cancer and we identified her to be a non-carrier of the *MUTYH* variant. Although no *MUTYH*-related cancers were reported for the parents of the proband, her maternal grandfather and female sibling were both affected with colorectal cancer at relatively young age, age 52 years and 39 years, respectively.

### 3.4. Tumor Sequencing to Assess Role of MUTYH Variant in a Suspected Familial EC Case

To explore the potential role of the germline heterozygous *MUTYH* variant in cancer development, we conducted tumor DNA sequencing of the heterozygous *MUTYH* variant carrier (case 28) to establish whether there was evidence of tumor variant enrichment and whether the *MUTYH*-associated mutational signature could be detected. We performed whole genome sequencing of an archival endometrial tumor block from the *MUTYH* carrier. The read depth was too low to accurately assess evidence of loss of heterozygosity at the *MUTYH* locus, although an increase in the percentage of variant reads from 43% in germline (16 of 37 reads) to 67% in the tumor (six of nine reads) was suggestive of tumor variant enrichment. The sequencing analysis also revealed a high proportion of C>T and C>A somatic mutations (Figure 3A). The pattern of C>T mutations is similar to COSMIC Signature 1, identified in many tumors and typically attributed to aging or deamination [48] and may be present due to formalin fixation. By performing signature assignment analysis we attributed 41% of all somatic single nucleotide variants to Signature 18 (Figure 3B), previously associated with inactivation of *MUTYH* in a series of familial colorectal cancer and adrenocortical carcinomas [49], indicating that the germline variant was driving the pattern of somatic mutations, and underlay development of EC in this individual.

## 4. Discussion

Based on the existing clinical management guidelines, a previous review suggested that only six genes currently have sufficient evidence of association with EC risk to be appropriate for hereditary EC diagnostic testing; these include the MMR genes (*MLH1*, *MSH2*, *MSH6* and *PMS2*), *EPCAM* (deletions due to their effect on *MSH2*) and *PTEN* [4]. We explored the role of candidate EC risk genes [4] beyond the MMR and *PTEN* genes, by analyzing an EC sample set unselected for family history and a cohort of familial EC cases. We also performed tumor sequencing analysis to explore whether these genes are cancer drivers associated with somatic mutagenesis in endometrial tumors. 

The findings suggest that variation in the following genes should be considered in future studies of EC risk: *ATM, MUTYH, PALB2, RAD51C* and *NBN*. *PALB2* was highlighted by both case–control and suspected familial EC analysis. Tumor mutational signatures provided evidence that germline variation in *BRCA1*, *PALB2*, *RAD51C, MUTYH* and *NTHL1* can be (but is not always) associated with tumor mutational signatures consistent with a functional role of these genes in endometrial tumor development. 

*ATM* encodes for a cell cycle checkpoint kinase that initiates DNA damage response via error-free repair pathway, HR, for double-stranded DNA breaks [50]. The *ATM* variant identified in a suspected familial EC case was classified as pathogenic for the rare autosomal recessive ataxia-telangiectasia syndrome by multiple submitters in ClinVar. The syndrome manifests a variety of phenotypic characteristics, including high incidence of cancer. Pathogenic variants in *ATM* are associated with increased breast cancer risk. Monoallelic c.7271T>G carriers are at a significantly increased risk, a 60% cumulative risk by age 80 years, similar to penetrance conferred by pathogenic germline variants in *BRCA2* [51]. Monoallelic carriers of other loss of function variants are reported to have a moderate increased risk of developing breast cancer (3-fold; 95% CI: 2.1–4.5) [52]. A number of *ATM* variants predicted to be deleterious to ATM protein function have been identified in EC cases, in unselected as well as a familial setting [7,53]. Another recent study [22] reporting results from germline panel testing of unselected EC cases identified *ATM* pathogenic variants as among the most common alterations observed (9/1170 cases), and estimated risk for *ATM* carriers to be OR 1.86 (*p* = 0.07) by comparison of case frequency to gnomAD non-Finnish European controls. Given that *ATM* loss of function variants are estimated to be associated with only a modest risk of breast cancer (OR 3.0, 95% CI 2.1–4.5) [52], larger well-designed studies will be required to determine if *ATM* variation confers a similar modest level of risk to EC.

*PALB2* encodes for one of the key proteins involved in the HR DNA damage repair by recruiting BRCA2 to DNA breaks [54]. The *PALB2* truncating variant identified in our familial cohort has been classified as a pathogenic variant for familial breast cancer by multiple submitters to ClinVar. *PALB2* is emerging as a gene that confers a high risk of breast cancer, with data suggesting individuals with pathogenic variants in *PALB2* have a high lifetime risk of around 32% [55]. *PALB2* variants have also been associated with increased risk of ovarian and pancreatic cancers [56]. In our study, the EC patient carrying the *PALB2* variant had a strong family history of various cancers, with carrier or obligate carrier status confirmed for relatives with breast, colon and EC. EC has been reported in relatives of breast cancer patients known to carry loss of function variants in *PALB2* [57], but carrier status was not confirmed. *PALB2* loss of function variants have also been detected in EC patients in several previous studies [45,57,58,59,60,61]. The results to date indicate that the role of *PALB2* loss of function variants in conferring EC risk should be further explored. 

Other HR pathway genes implicated in this study were *BRCA1*, *NBN* and *RAD51C*. Interestingly, while *NBN* and *RAD51C* had a more than 2-fold increased variant frequency in the EC sample set, *BRCA1* did not. To date, the role of *BRCA1* or *BRCA2* in EC risk has been much debated, with numerous conflicting reports [4]. Overall findings indicate that increased EC risk for *BRCA1/2* carriers has been associated with tamoxifen use for breast cancer prevention or treatment (since these genes confer high breast cancer risk) comparable to the risk observed in the general population [18,19]. There is also suggestive evidence that *BRCA1* pathogenic variants may confer a modest risk EC increase in the absence of tamoxifen exposure, particularly for serous and serous-like subtype cancers [62,63]. Unfortunately, the patient cancer history or tamoxifen exposure was not well documented for the TCGA-UCEC cancer cohort used in this study, hence we were unable to assess the possible contribution of tamoxifen for *BRCA1* or other genes that confer breast cancer risk. *RAD51C* has been recently shown to confer moderate risk for breast (relative risk (RR) = 1.99, 95% CI: 1.39–2.85) and high risk for ovarian cancers (RR = 7.55, 95% CI: 5.6–10.19) [64]; however, there have only been observational studies so far for EC [7,65]. Given the breast cancer risk, future studies on *RAD51C* and EC risk will need to account for tamoxifen exposure, same as for *BRCA1/2* genes. The role of *NBN* in EC risk has largely been unexplored. It is notable that while certain *NBN* variants have previously been reported to increase breast cancer risk [66], the most recent evidence from a large-scale case–control analysis refutes (OR 0.90, 95% CI 0.67–1.20) an association of truncating *NBN* variants with breast cancer risk [67]. 

In addition to considering a role of the above-mentioned HR-related genes in EC risk, we also investigated their potential role in EC development by analyzing tumor mutational signatures. We observed HR-associated mutational signature (Signature 3) in most tumors with *BRCA1*, *PALB2* and *RAD51C* pathogenic or likely pathogenic variants, but not in tumors with *BRIP1*, *CHEK2*, *FAN1* or *NBN* variants. This is consistent with previous reports in breast cancer and cell line experiments where Signature 3 was only detected for *BRCA1/2*, *PALB2* and *RAD51C* genes but not *ATM* or *CHEK2* [68,69]. The presence of Signature 3 in cases with *BRCA1*, *PALB2* and *RAD51C* variants, as well as tumor enrichment of these variants, suggest that these cancers are HR-deficient. Our observation is also supported by the report of tumor loss of heterozygosity in serous/serous-like EC with germline *BRCA1* mutations (two of three cases) [62]. 

Other genes implicated in this study included DNA base excision repair genes, *MUTYH* and *NTHL1*. Signature 36 (COSMIC v3), similar to Signature 18 detected in this study (COSMIC v2), has been associated with inactivation of *MUTYH* in MAP colorectal cancer [70] and observed in 5% of pancreatic neuroendocrine tumors that bore heterozygous germline *MUTYH* variants and subsequent loss of the wildtype allele in the tumor [71]. Together these observations indicate that oxidative DNA damage due to *MUTYH* inactivation may contribute to cancer etiology in several organs. In our study, a *MUTYH* variant considered pathogenic for MAP was likely enriched in the tumor of a suspected familial EC case, which presented with a tumor mutational signature consistent with the driver status of the *MUTYH* variant. However, *MUTYH* pathogenic variants were not more common in the TCGA unselected EC cohort relative to the population reference group (1.62% vs. 1.73%), and there was no evidence for tumor enrichment or appropriate tumor mutational signature in the TCGA cases. The majority of *MUTYH* pathogenic variants identified were two well recognized common pathogenic variants identified in the Western population to cause MAP (c.536A>G (p.Tyr179Cys), as detected in the suspected familial EC case; and c.1187G>A (p.Gly368Asp)) [47]. MAP is an autosomal recessively inherited predisposition to adenomatous polyposis and colorectal cancer [72]. The cumulative colorectal cancer risk to age 70 years for biallelic carriers is reported to be 75% (95% CI: 41–97%) for males and 72% (95% CI: 41–97%) for females, and for monoallelic carriers it is estimated to be 7% (95% CI: 5–11%) for males and 6% (95% CI: 4–9%) for females [73]. Indeed, the case reported here had a family history of colorectal cancer (in two relatives, ages 39 and 52). However, the risk of extracolonic cancers for *MUTYH* monoallelic pathogenic variant carriers with a family history of colorectal cancer is still uncertain, current evidence derived from a single study estimated cumulative risk of EC age 70 to be 4% (95% CI: 2–8%) for monoallelic *MUTYH* carriers [11], with an updated analysis of the same cohort [74] reporting a modest 2-fold EC risk for carriers (95% C.I 1.1–3.9). These previous findings suggest that, *MUTYH*-associated risk of EC, if validated, is likely to be extremely modest.

We have identified several genes in this study as possible additional EC risk genes; however, these results should be considered preliminary, and require further exploration in the follow-up studies. Although this study has not provided conclusive evidence regarding the role of the aforementioned genes in EC risk, results could nevertheless be of relevance as secondary findings for the patient and their relatives. We have shown that at least some germline carriers had a tumor mutational signature supportive of the driver role of the respective gene in cancer development. Overall, 28% of carriers of HR-related gene variants had a presence of HR-deficiency associated tumor mutational signature, which increased to 86% of carriers when only well-recognized HR genes (*BRCA1*, *PALB2* and *RAD51C*) were included [75]. This has implications for patient treatment decisions, since HR-deficient cancers are known to respond to PARP inhibitors [76]. Furthermore, cases with MMR-deficient or base excision repair-deficient (*MUTYH* or *NTHL1-*driven) tumors are likely to show hypermutated profiles, and thus would be good candidates for immunotherapy treatment, given the likely increase in neo-antigen production [77]. Our findings suggest potential value in secondary tumour profiling on identification of a germline gene alteration in EC patients, irrespective of a confirmed role of that gene in EC risk. Furthermore, somatic only changes would have the same implications for treatment decisions. It will thus be important to explore the overall proportion of EC cases with actionable tumor mutation profiles to determine the clinical value of unselected tumor mutational profiling.

## 5. Conclusions

We used genome sequencing and tumor mutational signature analysis to explore the role of purported EC risk genes in an EC sample set unselected for family history, and to identify candidate germline variants underlying the genetic cause of familial EC without MMR defects. Ancestry-matched case–control comparisons of germline variant frequency and/or sequence data from the suspected familial EC cases proposed several preliminary candidates for future risk association studies, with *PALB2* highlighted by both approaches. Tumor analysis highlighted germline variation in HR-related repair genes, particularly *BRCA1*, *PALB2* and *RAD51C,* to have a potential driver role in EC development based on the presence of mutational signature indicative of HR deficiency. For the heterozygous germline variants in other DNA damage repair genes, *MUTYH* and *NTHL1,* the mutational signature analysis indicates possible involvement in the etiology of EC, but only when there were indications of the germline variant being enriched in the tumor. 

Inclusion of these highlighted genes in clinical testing panels for EC predisposition will require results from further large-scale studies, to assess the level of EC risk associated with loss of function variation in these genes. Such studies should preferably follow a population-based case–control design and consider the role of other genetic and environmental factors in disease penetrance, including previous exposure to tamoxifen. While we anticipate that genes outside of MMR pathway are unlikely to explain a large component of suspected familial EC, our results indicate that additional tumor signature analysis for individuals with a germline gene alteration has potential to impact therapeutic decisions. 

## Figures and Tables

**Figure 1 cancers-13-01762-f001:**
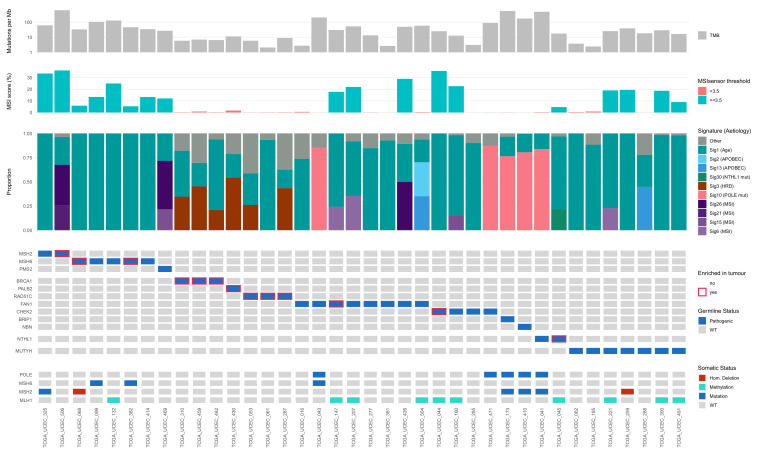
Somatic mutational signature analysis of the germline variant carriers in the TCGA-UCEC cohort. Tumor mutation burden (TMB), microsatellite instability (MSI) scores and mutational signatures observed in the TCGA-UCEC cases with pathogenic or likely pathogenic variants in DNA damage repair genes associated with specific mutational signatures.

**Figure 2 cancers-13-01762-f002:**
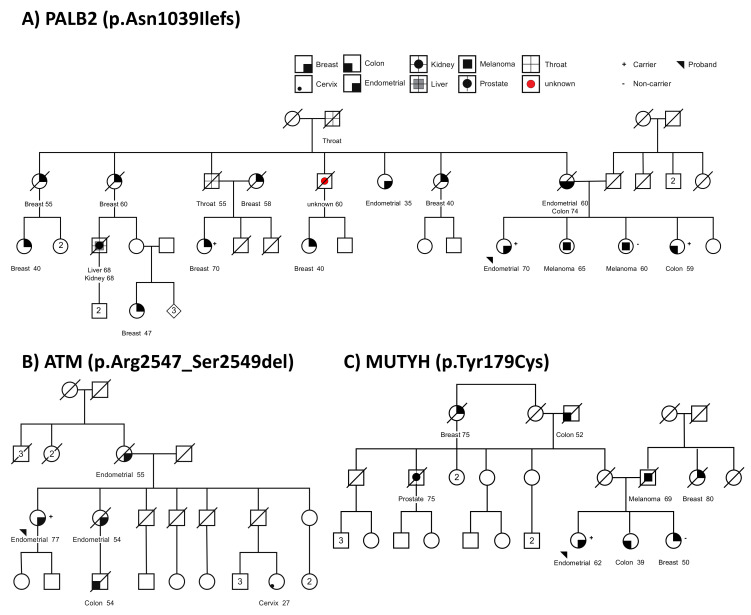
Pedigrees of families of the endometrial cancer cases carrying candidate variants. (**A**) Family pedigree of *PALB2* p.Asn1039Ilefs carrier. (**B**) Family pedigree of *ATM* p.Arg2547_Ser2549del carrier. (**C**) Family pedigree of the endometrial cancer case carrying candidate *MUTYH* p.Tyr179Cys. Squares symbolize males, circles symbolize females. Affected individuals are indicated by highlighted symbols, with cancer type and age at diagnosis noted below. Unaffected individuals are indicated by empty symbols. Endometrial cancer proband sequenced is indicated by black arrow below the symbol. Variant carriers are indicated by a (+) symbol and the non-carriers are indicated by a (−) symbol.

**Figure 3 cancers-13-01762-f003:**
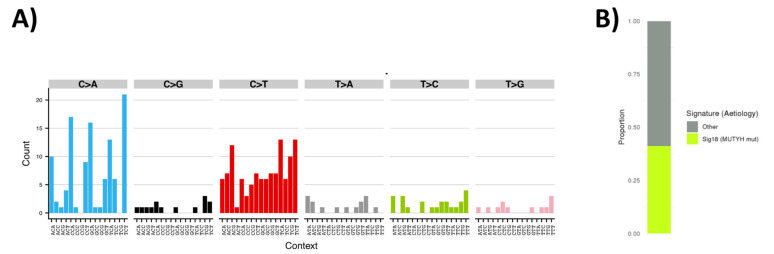
Somatic mutational signature analysis of the *MUTYH* germline variant carrier (suspected familial endometrial cancer cohort). (**A**) A total of 287 somatic single nucleotide variants (SNVs) identified in the endometrial tumor and used in signature analysis, plotted as counts in a 96 trinucleotide context. (**B**) The proportion of mutations in the tumor sample which were assigned to Signature 18.

**Table 1 cancers-13-01762-t001:** Overall frequency of pathogenic and likely pathogenic variants in 30 known and candidate endometrial cancer (EC) risk genes in an EC sample set (TCGA-UCEC study) and the general population (gnomAD).

Gene	Endometrial Cancer Cases				General Population			
	(TCGA-UCEC)				(gnomAD)			
	Number of Carriers	Number of Homozygote Carriers	Number of Total Cases	Carrier Frequency (%)	Number of Carriers	Number of Homozygote Carriers	Number of Total Cases	Carrier Frequency (%)
*MUTYH*	5	0	308	1.62	1023	3	59,095	1.73
***MSH6***	**4**	**0**	**308**	**1.3**	**134**	**0**	**59,095**	**0.23**
*CHEK2*	3	0	308	0.97	1099	7	59,093	1.86
***RAD51C***	**3**	**0**	**308**	**0.97**	**61**	**0**	**59,093**	**0.1**
*NTHL1*	2	0	308	0.65	268	0	59,090	0.45
***MSH2***	**2**	**0**	**308**	**0.65**	**11**	**0**	**59,092**	**0.02**
*SEC23B*	1	0	308	0.32	197	0	59,094	0.33
*FAN1*	1	0	308	0.32	186	0	59,095	0.31
*BRCA1*	1	0	308	0.32	140	0	59,095	0.24
***NBN***	**1**	**0**	**308**	**0.32**	**89**	**0**	**59,072**	**0.15**
***PALB2***	**1**	**0**	**308**	**0.32**	**85**	**0**	**59,094**	**0.14**
***PMS2***	**1**	**0**	**308**	**0.32**	**76**	**0**	**59,095**	**0.13**
*ATM*	0	0	0	0	284	0	59,088	0.48
*BRCA2*	0	0	0	0	182	0	59,079	0.31
*BRIP1*	0	0	0	0	123	0	59,090	0.21
*FANCC*	0	0	0	0	104	0	59,095	0.18
*RINT1*	0	0	0	0	55	0	59,094	0.09
*APC*	0	0	0	0	50	0	59,090	0.08
*MLH1*	0	0	0	0	34	0	59,095	0.06
*EPCAM*	0	0	0	0	32	0	59,092	0.05
*PTEN*	0	0	0	0	27	0	59,095	0.05
*SDHB*	0	0	0	0	20	0	59,089	0.03
*TP53*	0	0	0	0	20	0	59,095	0.03
*SDHC*	0	0	0	0	14	0	59,093	0.02
*SDHD*	0	0	0	0	7	0	59,095	0.01
*AKT1*	0	0	0	0	4	0	59,094	0.01
*PIK3CA*	0	0	0	0	3	0	58,839	0.01
*STK11*	0	0	0	0	2	0	58,753	0
*POLD1*	0	0	0	0	0	0	59,092	0
*POLE*	0	0	0	0	0	0	59,095	0

Only cases with Non-Finnish European ethnicity were included. Genes highlighted in bold had a frequency of >2 times higher in TCGA-UCEC compared with gnomAD. Carrier frequency represents the sum of all (likely) pathogenic variants in that gene. Genes highlighted in bold had more than double variant carrier frequency in the EC cases compared to the gnomAD population.

## Data Availability

TCGA-UCEC data were downloaded from GDC data portal in October 2016. GnomAD variant files (r.2.1.1) were downloaded from the gnomAD portal in April 2019. ANECS sequencing data are available upon reasonable request and subject to ethics approval.

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
