# Peer review of "Tumor Signature Analysis Implicates Hereditary Cancer Genes in Endometrial Cancer Development"

_cancers, 2021, doi:10.3390/cancers13081762_

Round 1

Reviewer 1 Report

I'm impressed with the work You've done.

I only missed the expalanation of some abbreviations i.e.: HR, ATM, MUTYH, ect.) and short descriptions of what are those responsible for.

Author Response

We thank the Reviewer for the kind comments.

HR is first defined on line 76.

We have updated the discussion to include short descriptions of the discussed genes and pathways (lines 356,375,422).

Reviewer 2 Report

this is a well written manuscript that contains useful information about the molecular characteristics of endometrial cancer

Author Response

We thank the Reviewer for noting that our manuscript provides useful information for molecular profiling of endometrial cancer. Reviewer required no additional changes.

Reviewer 3 Report

The paper is well written and well presented. The topic is largely treated. The authors described in depth the role of additional genes as explanation for familial EC presentation by investigating germline and EC tumor sequence data. They described the pathogenicity of variants found in several genes. Furthermore they performed an interesting tumor mutational signature analysis.

However, the authors have to clarify same points.

  1. Results reported in table 1 should be clarified: carrier frequency > of 1% could make the variant a polymorphism? Please explain this point to make the table results more clear.
  2. The authors in the table 1 caption and in the text named the variations as “presumed pathogenic”: please clarify is this variants are likely pathogenic or other.
  3. The study reported here, identified several additional genes for further exploration in relation to endometrial cancer risk and therapy and for this reason can be considered as “preliminary”, please clarify this point in the Discussion section, also by considering it as a limitation.
  4. What is the author’s opinion regarding the role of genes panels analysis in the EC screening in at-risk subjects and in general population? Please clarify this point in the discussion section.

Author Response

Point 1.  We thank the reviewer for suggesting to clarify the results. Table 1 reports combined per gene carrier frequencies of all detected pathogenic and likely pathogenic variants. We have updated the table legend and footnote to clarify.

Point 2. We have updated the methods description to clarify how variants were termed:

“For the gnomAD and TCGA-UCEC dataset analysis, only pathogenic or likely pathogenic ClinVar variants or predicted truncating variants (termed as likely pathogenic in this study) were considered.”

We have also updated the text throughout from “presumed pathogenic” to “likely pathogenic”.

Point 3. We thank the reviewer for the suggestion. As such we have updated the discussion and conclusions to reflect that some of the results are preliminary and require further exploration.

Point 4. We have added a sentence in the discussion addressing this point:

“Based on the existing clinical management guidelines, a previous review suggested that only six genes currently have sufficient evidence of association with EC risk to be appropriate for hereditary EC diagnostic testing; these include the MMR genes (MLH1, MSH2, MSH6 and PMS2), EPCAM (deletions due to their effect on MSH2) and PTEN [4].”